# Anti-Symmetric Electromagnetic Interactions’ Response in Electron Circular Dichroism and Chiral Origin of Periodic, Complementary Twisted Angle in Twisted Bilayer Graphene

**DOI:** 10.3390/molecules27196525

**Published:** 2022-10-02

**Authors:** Guoqiang Dai, Xiangtao Chen, Ying Jing, Jingang Wang

**Affiliations:** 1College of Science, Liaoning Petrochemical University, Fushun 113001, China; 2School of Physics, Northeast Normal University, Changchun 130024, China

**Keywords:** Z-TwBLG-QDs, anti-symmetric chirality, TEDM and TMDM, PDOS, topological analysis

## Abstract

Many novel physical properties of twisted bilayer graphene have been discovered and studied successively, but the physical mechanism of the chiral modulation of BLG by a twisted angle lacks theoretical research. In this work, the density functional theory, the wavefunction analysis of the excited state, and the quantum theory of atoms in molecules are used to calculate and analyze the anti-symmetric chiral characteristics of zigzag-edge twisted bilayer graphene quantum dots based on periodic complementary twisted angles. The analysis of the partial density of states shows that Moiré superlattices can effectively adjust the contribution of the atomic basis function of the fragment to the transition dipole moment. The topological analysis of electron density indicates that the Moiré superlattices structure can enhance the localization of the system, increasing the electron density of the Moiré central ring, reducing the electron surge capacity in general and inducing the reversed helical properties of the top and underlying graphene, which can be used as the origin of the chiral discrimination; it also reveals the mole in the superlattice chiral physical mechanism. On this basis, we will also study the nonlinear optical properties of twisted bilayer graphene based on a twisted angle.

## 1. Introduction

Twisted bilayer graphene (TwBLG) is a robust correlation system obtained by stacking coupling and interlayer rotation between one layer of graphene nanosheets and another layer of graphene nanosheets. This interlayer interaction will cause the symmetry of the electron to break [1]. For example, most Mott insulators exhibit an antiferromagnetic arrangement of electron spin [2,3]. In 2018, Pablo Jarrilo–Herrero et al. experimentally observed for the first time that magnetic angle graphene could achieve insulation and low-temperature superconductivity under strong electric field conditions slightly above absolute zero, which aroused considerable interest among researchers [4]. Defective bilayer graphene (BLG) with a small twisted angle will generate the early experimental preparation process [5]. The distortion angle causes the slight dislocation of the cellular lattice to produce a long period Moiré superlattice. This induces a periodic potential field and a microgap at the edge of the Briviine region of the Moiré superlattice (MS) [6]. As the interaction increases, the Quantum Anomalous Hall Effect (QAHE) and chiral edge states will occur near the insulating state [7]. The MS can be produced in a variety of ways, the latest being by cold-pressing graphite sheets [8]. The Moiré superlattice can effectively regulate the distribution of the magnetic transition dipole moment to control the chirality [9]. The MS structure has a circular dichroism due to its differential response to polarized light [10]. This optical activity originates from the relative rotation of the top and underlying layers, leading to the rotation of the helicity of Dirac Fermions [11,12]. The transition from a perfect tunnel reflection to a partial reflection by the height and energy of the barrier demonstrates the adjustable chirality of twisted bilayer graphene [13,14]. However, the physical mechanism related to the modulation of the twisted angle on the chirality of BLG has not been studied, in particular, circular dichroism with antisymmetric properties. The main purpose of this paper is to study the antisymmetric chirality of zigzag-edge TwBLG-QDs (Z-TwBLG-QDs) based on periodic complementary twist angles by first principles and wave function analysis, and to study the physical mechanism of the chirality of the mole in superlattices by the topological analysis of electron density.

We simulated the Z-TwBLG-QDs based on the periodic complementary twisted angle by precisely controlling the twisted angle between layers and the stacking mode of polar 2D graphene nanosheets [15,16,17,18]. One-photon absorption revealed the optical absorption properties of quantum dots. The visualizing electromagnetic interaction of the light-quantum dot system [19] is used to study the axisymmetric circular dichroism in ECD. Atomic orbital transitions of DOS reveal the anti-symmetric distribution of the transition electric dipole moment (TEDM) and the transition magnetic dipole moment (TMDM). A topological analysis of electron density reveals the physical mechanism by which MS determines chirality.

The graphene monolayer is a regular hexagon structure, and the edge is composed of seven benzene rings. We use the modeling software MS to adjust the two graphene nanosheets to the appropriate layer spacing, and through the Van der Waals force between layers, the micro nano “Van der Waals superposition structure” quantum dots are formed. This structure makes BLG exhibit a strong interaction between coupling layers [20]. Keep the geometric center of the graphene nanosheet as the origin, and rotate the top graphene nanosheet through a specific “magic angle”. When the twisted angles are 27.8° and 32.2°, the periodic complimentary relationship (the sum of angles are 60°, and the Moiré superlattice is the same during the rotation of every 60° of the top layer graphene) is satisfied. Two structures of MS with the same shape but different rotation directions (Z-TwBLG-QDs-27.8° and Z-TwBLG-QDs-32.2°) are obtained, see Figure 1a,b.

## 2. Results

The UV–vis and ECD spectra are calculated to study the optical absorption characteristics of quantum dot system. As shown in Figure 1c, the spectra of Z-TwBLG-QDs-27.8° and Z-TwBLG-QDs-32.2° overlap. The main absorption peak is near 772.3 nm, which is contributed to by S_15_, and the second absorption peak is contributed to by two excited states (S_49_ and S_53_). The main absorption peak of Z-TwBLG-QDs-32.2° is near 772.4 nm, which is contributed to by two excited states (S_15_ and S_16_), and the secondary peak is contributed to by two excited states (S_49_ and S_53_). As shown in Figure 1d, ECD shows two regions with high rotatory strength in the visible light region. When the twisted angle is 27.8°, the corresponding ECD excited states are S_51_/S_53_ and S_47_/S_49_. When the twisted angle is 32.2°, the corresponding excited states are S_51_/S_53_ and S_48_/S_49_. The excited states corresponding to the absorption peaks in the near-infrared region are S_13_ and S_14_. In addition, the rotatory intensity of the excited states of the two systems at the same wavelength is axisymmetric, and the absolute value is almost the same (see Appendix A). Therefore, the electromagnetic interaction intensity of their incident light-quantum dots is very close. The symmetrical rotatory strength makes the molar absorption coefficients of Z-TwBLG-QDs-27.8° and Z-TwBLG-QDs-32.2° satisfy an axisymmetric relationship, indicating that the molecular system exhibits symmetric circular dichroism in response to polarized light.

Circular dichroism is the asymmetric response of the electric and magnetic transition dipole moments when the system interacts with electromagnetic waves, which can reflect the chiral properties of the system. The TEDM and TMDM are plotted to reveal the physical mechanism of the axial symmetry of the ECD spectrum. The transition dipole moment density is the product of the transition density and the coordinate variables in the three directions x, y, and z of the cartesian coordinates [21]:(1)$$T_{k}\left(r\right)=-kT\left(r\right),k=xyz$$

The transition dipole moment density here is connected to the transition electric dipole moment, specifically. The contribution of the x, y, and z components to TEDM is obtained. The component of the transition magnetic dipole moment can be defined as
(2)$$M_{l}=\langle \Phi^{0}\left|m\frac{\partial }{\partial n}-n\frac{\partial }{\partial m}\right|\Psi^{exc}\rangle =\underset{i\to a}{\sum }w_{i}^{a}\langle \varphi_{i}\left|m\frac{\partial }{\partial n}-n\frac{\partial }{\partial m}\right|\varphi_{a}\rangle -\underset{j\leftarrow b}{\sum }w{’}_{j}^{b}\langle \varphi_{j}\left|m\frac{\partial }{\partial n}-n\frac{\partial }{\partial m}\right|\varphi_{b}\rangle ,lmn=xyz$$

The component of the transition magnetic dipole moment $m_{l}\left(r\right)$ is defined to visualize the transition magnetic dipole moment. The explicit expression is
(3)$$m_{l}\left(r\right)=\underset{i\to a}{\sum }w_{i}^{a}\varphi_{i}\left(r\right)\left[m\frac{\partial \varphi_{a}}{\partial n}\left(r\right)-n\frac{\partial \varphi_{a}}{\partial m}\left(r\right)\right]-\underset{j\leftarrow b}{\sum }w_{j}^{b}\varphi_{j}\left(r\right)\left[m\frac{\partial \varphi_{b}}{\partial n}\left(r\right)-n\frac{\partial \varphi_{b}}{\partial m}\left(r\right)\right],lmn=xyz$$

Its full-space integral is the transition magnetic dipole moment as follows:(4)$$M_{k}=\int m_{k}\left(r\right)dr,k=xyz$$

To make the distribution characteristics of TEDM and TMDM more precise and more intuitive, the smoothing electric/magnetic moments (TEDM_S_ and TMDM_S_) of the system on the right side of TEDM and TMDM in Figure 2, Figure 3 and Figure 4 are performed. TEDMS and TMDM_S_ are equivalent to describing the distribution of TEDM and TMDM with Gaussian functions equivalently, and the distribution details are erased. At this time, the isosurface map of TEDM and TMDM becomes an ellipse. When the twisted angle is 27.8°, the TEDM of S_13_ at 844.5 nm is symmetrically distributed in the x, y, and z components. The isosurface is gathered at the edge of the BLG, which corresponds to the long axis of the smoothing isosurface of the ellipse. The positive and negative isosurfaces of TEDM on the z component are wholly separated. The TMDM is distributed on the axis of symmetry on the x and y components. The distribution is uniform in the z component and tends to the Moiré superlattice, see Figure 2a–c. S_14_ is the degenerate state of S_13_, and their TEDM and TMDM on the x and y components are perpendicular to each other. ECD shows that the rotatory intensities of the two excited states contribute negatively to ECD. When the twisted angle changes to 32.2°, the absorption peak at the same wavelength is contributed to by S_13_. Compared with S_14_, the TEDM and TMDM on the x, y, and z components are anti-symmetric, so S_13_ is responsible for ECD absorption. S_14_ is responsible for chirality.

Comparing the ECD excited state of the two systems at 844.5 nm, the distribution range of TEDM_S_ in S_13_ is more extensive in the x component, and the TEDM of S_14_ is more extensive in the y component, see Figure 2d,e,g,h. On the z component, the TEDM and TMDM distributions are also anti-symmetric. The TEDM_S_ and TMDM_S_ show that the positive and negative isosurface face have opposite directions, see Figure 2f,i, which indicates that the axisymmetric ECD spectrum at 844.5 nm is mainly contributed to by the anti-symmetric distribution of the TEDM in the z component. The excited states of Z-TwBLG-QDs-27.8° and Z-TwBLG-QDs-32.2° of chiral reversal at 614.5 nm are S_47_ and S_48_, respectively. Their sizes of the TEDM and TMDM isosurface in Cartesian coordinates and distributions are very close, so these two excited states are responsible for optical absorption in the ECD, see Figure 3a–f.

The ECD excited state of the two systems at 612.8 nm is contributed to by S_49_. When the twisted angle is 27.8°, the TEDMs on the x and y components are distributed axisymmetrically on the edge of the BLG, TEDM_S,_ and TMDM_S_ shows that the negative value is greater than the positive value, the TEDM isosurface on the z component is significantly reduced, and the TEDM_S_ shows the more significant degree of separation of positive and negative values. The isosurface distribution of TMDM also has axial symmetry. The proportions of positive and negative values on the x and the y component are similar, and the positive and negative values on the z component are separated to a greater extent, see Figure 3g–i. When the twisted angle is 32.2°, the negative value of TEDM on the x component is greater than the positive value, and the situation on the y component is the opposite. Additionally, the positive–negative value ratio of TMDM changes. On the z component, the positive and negative isosurfaces also separate to a greater extent and the directions are opposite, see Figure 4j–l. Therefore, the axisymmetric ECD of the two systems at 612.7 nm is mainly caused by the anti-symmetric distribution of the positive and negative isosurfaces on the z component.

The ECD excited state of Z-TwBLG-QDs-27.8° at 602.2 nm is S_51_. TEDM and TMDM are axisymmetrically distributed in the x and y components. The TEDM in the z component is significantly reduced, but the TMDM is significantly increased. The edge distribution spreads to the Moiré superlattice, and the positive and negative values of the TEDM_S_ and TMDM_S_ are more evenly distributed, see Figure 4a–c. When the twisted angle is 32.2°, TEDM and TMDM have the same changing trend. The difference is that the ratio of the positive and negative values of the TMDM_S_ on the x component changes, the TEDM on the y component changes, and the orientation of TEDM_S_ on the z component is opposite, see Figure 4d–f. Another group of ECD excited states near 598.2 nm is S_53_. The distribution characteristics of TEDM and TMDM are similar to S_51_. The TMDM_S_ of Z-TwBLG-QDs-27.8° on the x component are larger, and the negative value of TEDM_S_ on the y component is larger, see Figure 4g–h, while for Z-TwBLG-QDs-32.2°, the negative value of the TEDM_S_ on the x component is larger, the positive value of TMDM on the y component is more significant, and the direction of TMDM_S_ on the z component is opposite, see Figure 4j–l. This situation is widespread in other ECD excited states with axisymmetric circular dichroism, and it intuitively reveals the physical mechanism of the anti-symmetric distribution of the transition electric dipole moment and the transition magnetic dipole moment.

The above analysis discusses the electromagnetic interaction mechanism of Z-TwBLG-QDs based on the periodic complementary twisted angles. It is necessary to investigate the DOS analysis of the area with anti-symmetric distribution of TEDM and TMDM to reveal the physical mechanism of the anti-symmetric distribution of TEDM and TMDM, corresponding to fragments 1–5, see Figure 5a,b. Fragment definition adds the basis functions on the atom to the fragment, which is marked with different colors. As shown in Figure 5c,d, the normalized integral curve of PDOS represents the contribution of each fragment’s atomic orbital, the relative positions of the blue curve, and the yellow curve alternate near the energy of −2.77 eV. According to Mulliken division, the contribution of the r-th basis function to the transition dipole moment is as follows:(5)$$d_{r}=\underset{s}{\sum }P_{r,s}^{tran}\langle \chi_{r}\left|-r\right|\chi_{s}\rangle$$
where $P^{tran}$ is the transition density matrix between certain two states and $\langle \chi_{r}\left|-r\right|\chi_{s}\rangle$ is the electric dipole moment integral between the r−th and s−th basis functions.

The contribution of the atoms to the transition dipole moment is to sum the basis function contributions of all atoms since the transition electric dipole moment is the transition dipole moment from the ground state to the excited state. Therefore, the alternating phenomenon of the contribution of the fragment atomic orbits will cause significant changes in the response mode of the transition electric dipole during the light-quantum dot interaction of the system. Combining Formulas (1) and (2), alternating atomic orbital contributions will change the orbital wavefunctions $\varphi_{a}$ and $\varphi_{b}$, thereby causing the changes in the transition electric dipole moment and the transition magnetic dipole moment. These changes will indicate the anti-symmetric distribution of TEDM and TMDM for the two systems. The MS structure that is based on the periodic complementary twisted angles on the atomic orbitals of the edge region are the cause of this situation.

The above discussion qualitatively reveals the physical mechanism of the axisymmetric ECD of the quantum dot system. To investigate the origin of the symmetric circular dichroism quantitatively, it is necessary to calculate the helical properties of chiral interactions in molecular systems. These can be investigated by the polarization (the sensitivity of electron density to deformation in an electric field) of atomic fragments in chiral molecules according to the theory of molecular polarizabilities [21]. The electron density is defined as
(6)$$\;\rho \left(r\right)=\underset{i}{\sum }\eta_{i}{\left|\varphi_{i}\left(r\right)\right|}^{2}=\underset{i}{\sum }\eta_{i}\left|\underset{\mu }{\sum }C_{\mu ,j}\chi_{\mu }\left(r\right)\right|$$
where $\eta_{i}$ is the occupied number of orbital i, φ is the orbital wavefunction, χ is the basis function, and C is the coefficient matrix. The element in the i−th row and j−th column correspond to the expansion coefficient of orbit j relative to the basis function i. When the twisted angle is 27.8°, the MS structure contains an axisymmetrically distributed Moiré superlattice along the central benzene ring of the graphene layer. The RCP of the Moiré superlattice is found in the chemical bond formed by the benzene ring on the top and underlying graphene.

As shown in Appendix A, the real space function of the RCP of the Coronene and the two Z-TwBLG-QDs are calculated through the topological analysis of the electron density. The $\rho_{RCP}$ of the two systems are very close. All RCP are numbered to analyze the correlation between the $\rho_{RCP}$ and the chirality of the system. The red serial numbers correspond to the RCP of the Moiré superlattice in the top graphene, and the blue serial numbers correspond to the RCP of the Moiré superlattice in the underlying graphene. Figure 6a indicates the changing trend of the normalized $\rho_{RCP}$. The polarization of the Moiré superlattice increases as the electron density increases after removing an RCP with the smallest polarization. The red circular arrow on the top layer of the graphene is the reduced order of the electron density, see Figure 6c. The rotation direction is clockwise, so the top layer graphene is a right-handed spiral, while the underlying graphene is a left-handed spiral, see Figure 6b,d. The Laplace function of the electron density ($\nabla_{\rho_{RCP}}^{2}$) is to calculate the gradient of $\rho_{RCP}$:(7)$$\nabla^{2}\rho \left(r\right)=\frac{\partial \rho \left(r\right)}{\partial x^{2}}+\frac{\partial \rho \left(r\right)}{\partial y^{2}}+\frac{\partial \rho \left(r\right)}{\partial z^{2}}$$

The normalized $\nabla_{\rho_{RCP}}^{2}$ is the same as the curve of normalized $\rho_{RCP}$ because $\nabla_{\rho_{RCP}}^{2}$ is the second derivative of $\rho_{RCP}$, see Appendix A. The numerical value indicates that $\nabla_{\rho_{RCP}}^{2}$ is positive, indicating that the interlayer interaction originates from the closed-shell interaction of the π-π bond in the Moiré superlattice. The interaction that is different from the covalent bond comes from the shared electron pair, so there is no electron accumulation in the bonding area. The increase in the degree of aggregation will reduce the surging ability of electrons and weaken the role of electrons in the electrostatic interaction of the region of MS, making the nuclear electrostatic potential of Z-TwBLG-QDs relative to Coronene increase, see Figure 6e,f.

The values of $\rho_{RCP}$ between the Coronene and the two Z-TwBLG-QDs reflect that the MS will enhance the localization. It is necessary to analyze the localized orbital locator (LOL) of the Coronene and the two systems of Z-TwBLG-QDs to further reveal the influence of the MS on the localization of the system. Appendix A show that the value of LOL at the central RCP of the two systems is significantly greater than that of Coronene. The color-filled map of LOL-pi at 1.2 Bohr above the plane of the molecular system was drawn to visually analyze this difference. As shown in Figure 7c, the LOL-pi coloring diagram of Coronene shows that the difference from RCP is that the outer-shell electrons of the carbon atom of the central six-membered ring have no significant changes due to the stable conjugated structure.

On the contrary, the π electrons delocalize between adjacent carbon atoms at the six protruding points of the zigzag edge. After adding the Moiré, the delocalization path of the π electron does not include the six-cornered outer benzene ring. Additionally, the delocalization gradually weakens from the outside to the inside because the gradient of orbital wavefunction of the system at the boundary is larger than the inside, see Figure 7d,e.

Another tool for analyzing the delocalization of electrons is the electron localization function (ELF), which has a similar physical meaning to LOL. The changing trend of the ELF value of the three systems’ RCP is the same as that of LOL, see Figure 7b. The previous discussion indicates that the electrons at the edge of Z-TwBLG-QDs are more active than the internal ones under the edge effect. The ELF_pi isosurface can intuitively reflect the delocalization characteristics of the molecular system contributed to by π electrons, see Figure 7f. When the pi isovalue of the six-membered ring in the Coronene center is broken, the isovalue is 0.495. When the pi isovalue of the six-member ring at the edge is broken, the isovalue is 0.518. Therefore, the conjugation degree of the central six-membered ring is lower than the edge, see Appendix A. When the isovalue is set to 0.7, it can be clearly seen that the π isosurface at the edge is more significant than the internal one, and this law also applies to the more extensive system such as Z-TwBLG-QDs. As shown in Figure 7g,h, the π isosurface of Z-TwBLG-QDs-27.8 gradually decreases from the edge to the internal, and the degree of π conjugation decreases. This situation is the same when the twisted angle is 32.2°.

## 3. Methods

The quantitative calculation of the two molecular systems consists of two steps. The first step is using Gaussian16 [22] combined with GFN2-xTB [23] to optimize the ground state geometry, and the GFN2 XTB method is similar to the semi empirical DFT of DFTB, with good accuracy and universality. The further optimization of the ground state structure uses the DFT method [24], B3LYP functional [25], the 6-31G(d) basis set [26], and DFT-D3 dispersion-correction [27]. The step of excited state calculation is to calculate the Boltzmann distribution of the model with internal energy, and the final structure is calculated by time-dependent density functional (TDDFT). The output of this step is used to calculate and visualize the electron-hole pair density, the transition density matrix (TDM), and all the configuration coefficients of the TMDM and TEDM. The wavefunction of excited state generates the electron-hole pair density through the linear combination of the single excitation configuration function [28]. TEDM and TMDM are determined by the tangential vector of the transition density space [29,30]. The molecular orbitals of the PDOS are calculated by the Hartree–Fock equation theoretical method, which is based on a single-electron approximation [31]. The topological analysis of electron density is based on the QTAIM [32], and the real space function (a function with three-dimensional space coordinates as variables) of the ring critical point (RCP) of the Moiré superlattice was obtained.

## 4. Conclusions

In this work, various first-principle calculation methods are used to analyze the chiral physical mechanism of Z-TwBLG-QDs. The analysis of partial density of states indicates that the MS based on the periodic complementary twisted angle will change the contribution of the basis functions of the edge atoms of the system to the transition dipole moment. That will cause the orbital wavefunction to change and induce an anti-symmetric transition electric dipole moment and a transition magnetic dipole moment, resulting in axisymmetric circular dichroism in ECD. The topological analysis of electron density shows that the chirality is caused by the opposite helical characteristics of top graphene nanosheets and the underlying graphene nanosheets of the Moiré superlattices system. The structure of the Moiré superlattices weakens the conjugation degree of the edge π electrons, which has a positive effect on the localization of the molecular system. This research is of great significance in revealing the chirality generation and regulation mechanism of layered two-dimensional chiral materials and elucidating the structure-function relationship of chiral molecules.

## Figures and Tables

**Figure 1 molecules-27-06525-f001:**
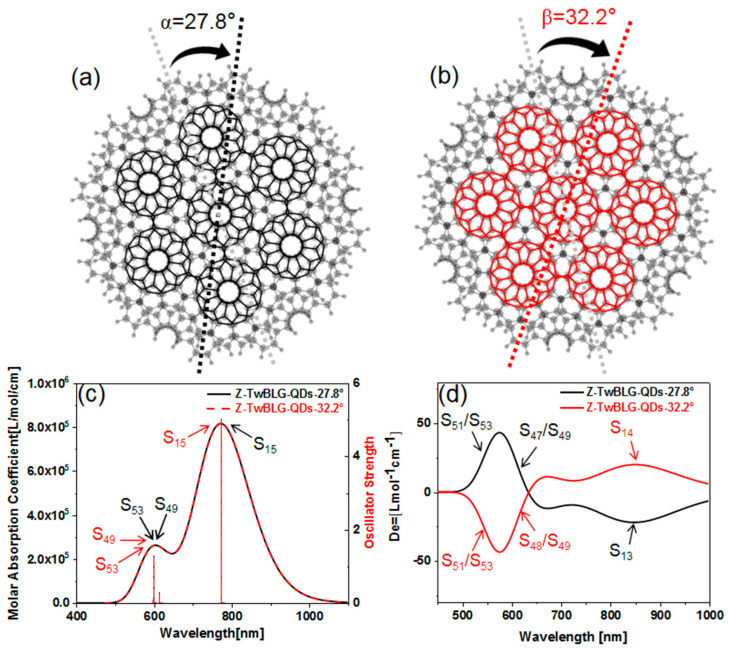
Molecular structures of Z-TwBLG-QDs-27.8° and Z-TwBLG-QDs-32.2° (**a**,**b**). The theoretical calculation of combined UV–vis and ECD of Z-TwBLG-QDs-27.8° and Z-TwBLG-QDs-32.2° (**c**,**d**).

**Figure 2 molecules-27-06525-f002:**
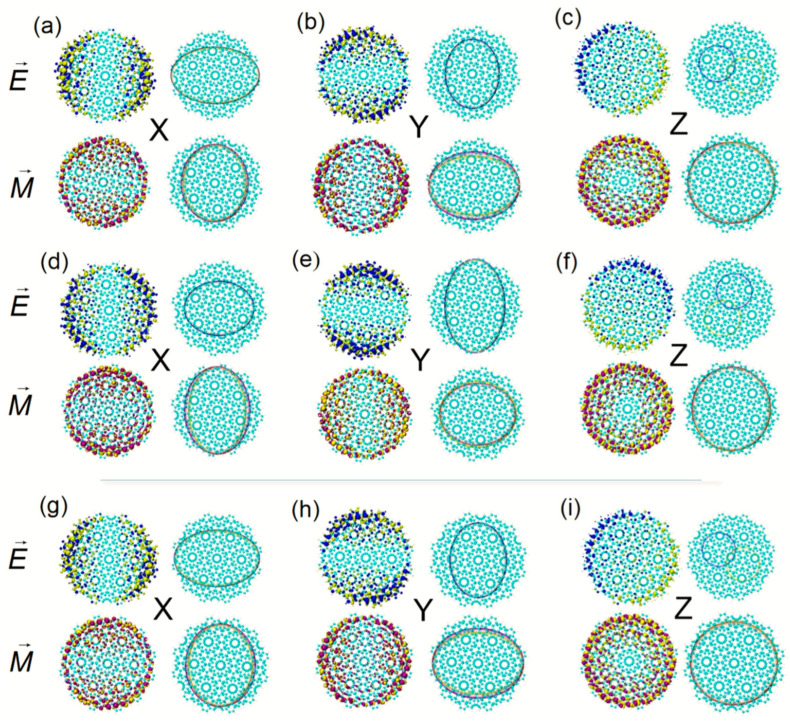
The theoretical calculation of TEDM (yellow isosurface represents a negative value, blue isosurface represents positive value) and TMDM (yellow isosurface represents a negative value, purple isosurface represents positive value) of Z-TwBLG-QDs-27.8° at S_13_ and its TEDM_S_ and TMDM_S_ of Cartesian component (**a**–**c**), respectively. TEDM and TMDM of Z-TwBLG-QDs-27.8° at S_14_ and its TEDM_S_ and TMDM_S_ of Cartesian component (**d**–**f**), respectively. TEDM and TMDM of Z-TwBLG-QDs-32.2° at S_13_ and its TEDM_S_ and TMDM_S_ of Cartesian component (**g**–**i**), respectively.

**Figure 3 molecules-27-06525-f003:**
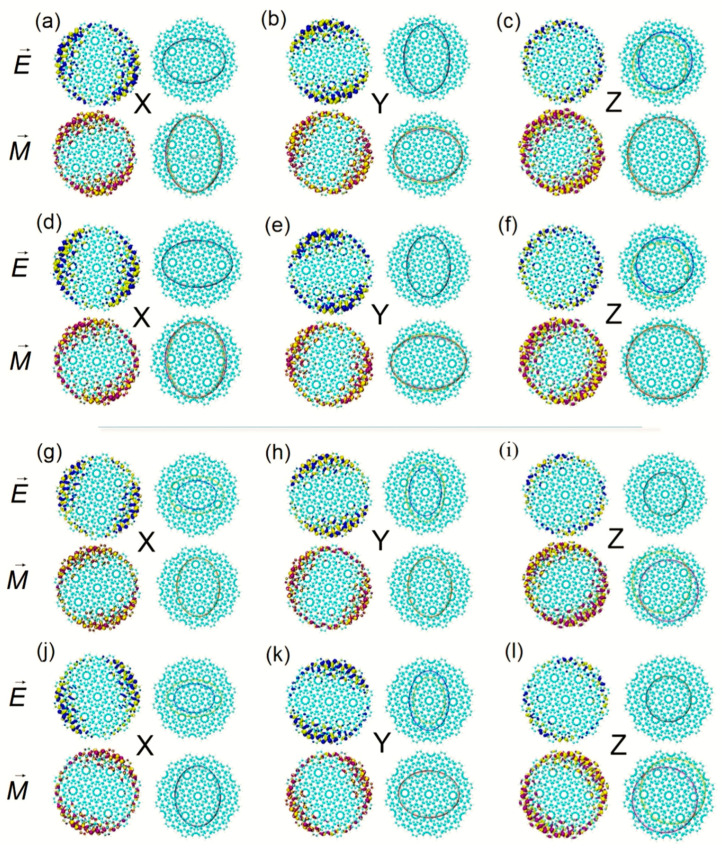
The theoretical calculation of TEDM (yellow isosurface represents a negative value, blue isosurface represents positive value) and TMDM (yellow isosurface represents a negative value, purple isosurface represents positive value) of Z-TwBLG-QDs-27.8° at S_47_ and its TEDM_S_ and TMDM_S_ of Cartesian components (**a**–**c**), respectively. TEDM and TMDM of Z-TwBLG-QDs-32.2° at S_48_ and its TEDM_S_ and TMDM_S_ of Cartesian components (**d**–**f**), respectively. TEDM and TMDM of Z-TwBLG-QDs-27.8° and Z-TwBLG-QDs-32.2° at S_49,_ and their TEDM_S_ and TMDM_S_ of Cartesian components (**g**–**i**,**j**–**l**), respectively.

**Figure 4 molecules-27-06525-f004:**
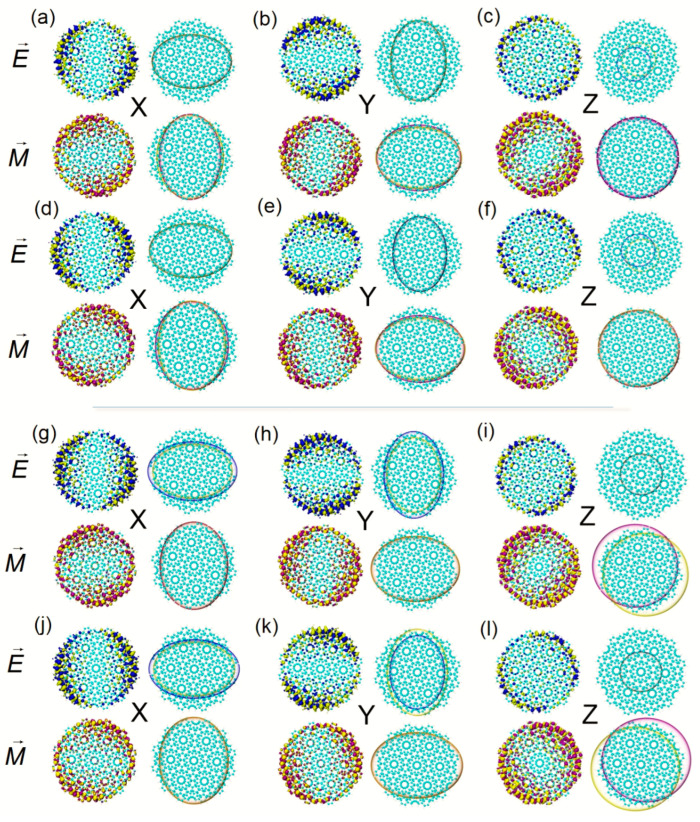
The theoretical calculation of TEDM (yellow isosurface represents a negative value, blue isosurface represents positive value) and TMDM (yellow isosurface represents a negative value, purple isosurface represents positive value) of Z-TwBLG-QDs-27.8° and Z-TwBLG-QDs-32.2° at S_51_ and their TEDM_S_ and TMDM_S_ of Cartesian components (**a**–**c**,**d**–**f**), respectively. TEDM and TMDM of Z-TwBLG-QDs-27.8° and Z-TwBLG-QDs-32.2°, and their TEDM_S_ and TMDM_S_ of Cartesian components at S_53_ (**g**–**i**,**j**–**l**), respectively.

**Figure 5 molecules-27-06525-f005:**
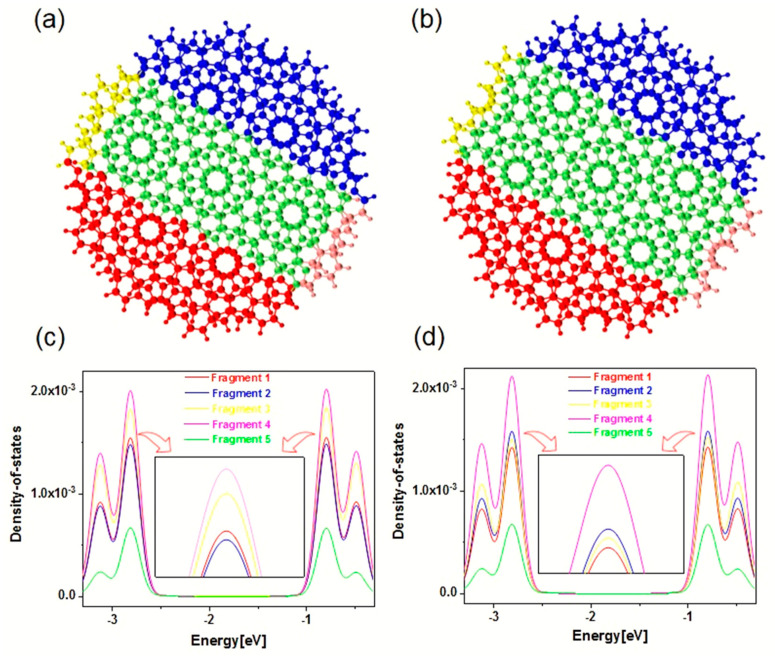
The theoretical calculation of atomic fragment definition of PDOS (**a**,**b**). The normalized PDOS which is divided by the number of atoms in the fragment integral curve of Z-TwBLG-QDs-27.8° and Z-TwBLG-QDs-32.2° (**c**,**d**).

**Figure 6 molecules-27-06525-f006:**
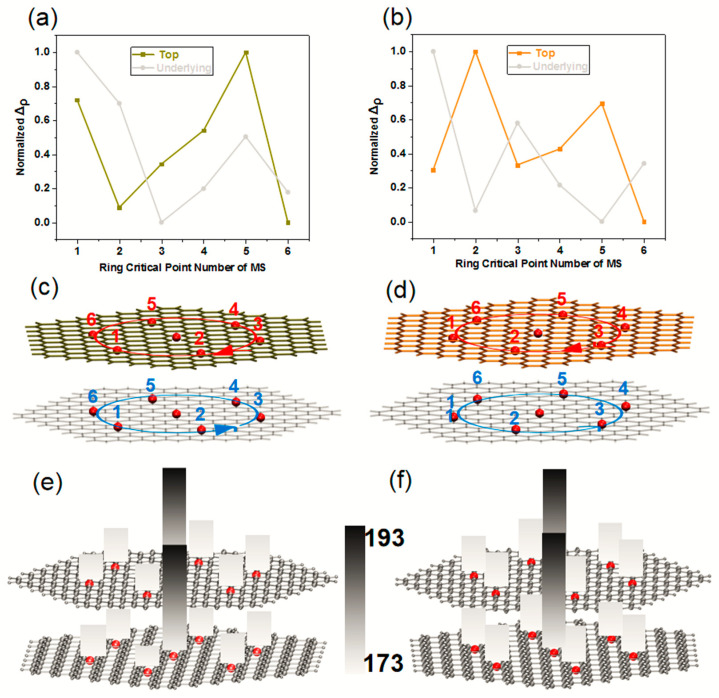
The theoretical calculation of normalized integral curves of $\rho_{RCP}$ in Moire superlattice RCP of Z-TwBLG-QDs-27.8° and Z-TwBLG-QDs-32.2° (**a**,**b**). The position of RCP in Z-TwBLG-QDs-27.8° and Z-TwBLG-QDs-32.2° in the Moiré superlattice; the circular arrow represents the direction of $\rho_{RCP}$ decreased (**c**,**d**). The theoretical calculation of nuclear electrostatic potential of RCP in Z-TwBLG-QDs-27.8° and Z-TwBLG-QDs-32.2° in the Moiré superlattice (**e**,**f**).

**Figure 7 molecules-27-06525-f007:**
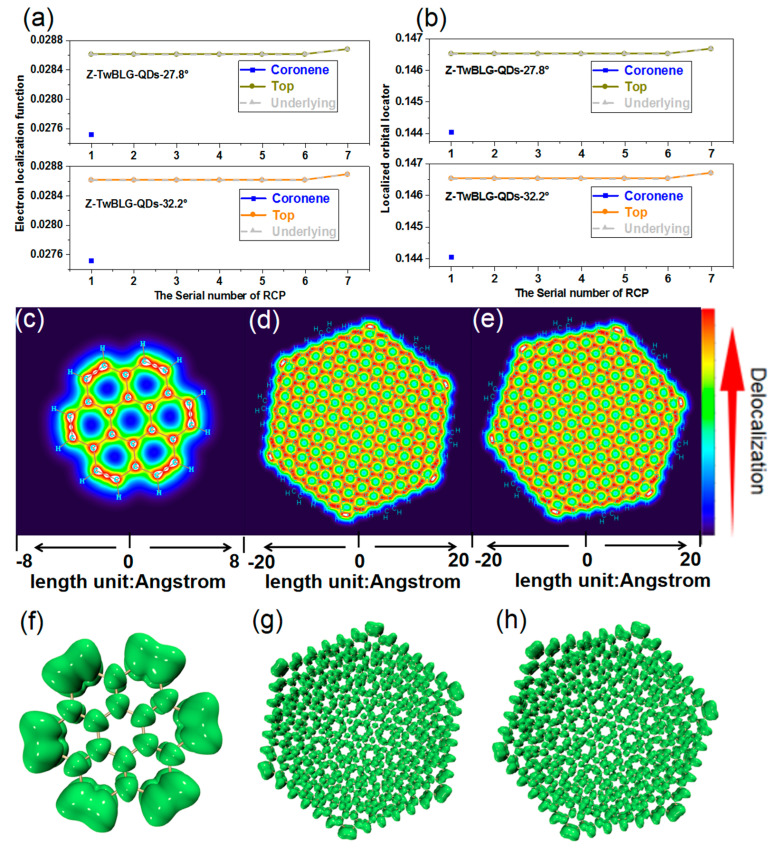
The theoretical calculation of LOL and ELF of RCP in Z-TwBLG-QDs-27.8° and Z-TwBLG-QDs-32.2° in the Moiré superlattice (**a**,**b**). LOL_pi color-filled map of Coronene, Z-TwBLG-QDs-27.8° and Z-TwBLG-QDs-32.2° (**c**–**e**). The ELF_pi isosurface of Coronene, Z-TwBLG-QDs-27.8°, and Z-TwBLG-QDs-32.2° (**f**–**h**); the isovalue is 0.7.

## Data Availability

Not applicable.

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
