# Peer review of "Anti-Symmetric Electromagnetic Interactions’ Response in Electron Circular Dichroism and Chiral Origin of Periodic, Complementary Twisted Angle in Twisted Bilayer Graphene"

_molecules, 2022, doi:10.3390/molecules27196525_

Round 1
Reviewer 1 Report
The work is original and of good quality. The manuscript should be published after a minor revision. Detailed feedback:
- The work might could be more reader friendly, the writing is solely purposed for physicist. It might be useful to have a summarizing table to guide the reader throughout the introduction. In a few words, what is the purpose of the study.
- In the introduction it might be useful to mention about the different approaches to generate Moire fringes, a recent one is the cold pressing of graphite flakes https://doi.org/10.1016/j.apmt.2022.101594
- Make clear in the figure captions weather the plots are experimental or calculated.
- The English should be polished, avoid repetition of words. Highlight key features in the figures. Add scale bars to the figures whenever appropriate.
- Reduce the number of abbreviations in caption, abstract and conclusions.
Author Response
Dear Professor
Thank you for your comment.
We have summarized the research purpose of the paper in the introduction section, and the additional parts are marked in red (page 6 in line 46 to 49).
Thank you very much.
Reviewer 2 Report
This is a computational modelling paper describing the interested properties of a simulated twisted Bilayer Graphene molecular system. Obviously, the initial materials concept originates in some interesting experimental observations. These should be cited and described in the introduction. And in conclusions the authors should indicate, how their results impact on experiments and could be verifies experimentally. Furthermore, in my opinion, there is too much detail in Figs 2-4. Much of these results can be included in a supplementary results file and only the most important data should be shown in the main text.
Reviewer 3 Report
Before making a final decision on this manuscript, I will give the authors an opportunity to make major revisions, which I list below:
Comment 1:
This title does not seem to be very sensitive, thus it needs to be rephrased in accordance with the manuscript's content.
Comment 2:
Parts of the manuscript were not free from grammatical and typographical errors to the extent that the authors began their manuscript with an obvious error in the abstract (In this works???). The language of the manuscript should be revised so that the manuscript is not rejected in the second round of reviews.
Comment 3:
The introduction is poorly written and should be reconsidered for improvement.
Comment 4:
The authors did not make a clear paragraph about novelty in the manuscript. The novelty of the manuscript should be clearly stated in the introduction.
Comment 5:
The references of the type [1-4], [13-15], [16-17] and [19-21], in the introduction are not justified. Authors should highlight the importance of their content.
Comment 6:
The quality of the figures should be improved.
Comment 7:
Please paraphrase the final conclusion, focusing on the main results of the work. The final conclusion should not be written as an essay.
In summary, I hope the authors will respond and address all the issues raised above so that I can make them happy with a positive decision in the second round.
Reviewer 4 Report
In the present work, the authors used the density functional theory (DFT) and the Quantum Theory of Atoms in Molecules (QTAIM) to calculate and analyze 11 the anti-symmetric chiral characteristics of zigzag edge twisted bilayer graphene quantum dots (Z-12 TwBLG-QDs) based on periodic complementary twisted angles. They found that Moiré superlattices structure can effectively adjust the contribution of the 14 atomic basis function of the fragment to the transition dipole moment. I found this paper valuable for the journal readers just after some minor revisions:
1- The necessity of doing this work should be stated clearly in the first sentence of the abstract section.
2- The future outline should be presented in the last sentence of the abstract section.
3- Don't use bulk references in the introduction section.
4- The introduction section is written poorly. I strongly suggest to authors to bold the necessity of doing their work in this section. Moreover, they need to litreture review in detail.
5- The "Two layers of graphene nanosheets are stacked together by interlayer van der Waals forces" needs a reference.
6- The computational section is written poorly. There is no information about the type of approximation, optimized k-points, cutoff, or other details about the computational method.
7- All the used equations need a reference.
8- The previous studies should validate the main outcome of this paper.
Round 2
Reviewer 3 Report
The authors have answered all of the issues raised. The revised manuscript is now acceptable for publishing, but there is one error in the abstract that I leave to the editor to report to the authors. In these works or In this work, NOT In this work(s).